# Image segmentation of cervical grainy sandy patches lesions associated with female genital schistosomiasis using deep convolutional neural network with U-NET architecture

Karl Emil Jøker[1]*, Peter Christian Derek Leutscher[1,2], Kristine Brøndbjerg Øby[1], Karoline Jøker[1], Bodo Sahondra Randrianasolo[3], Maciej Plocharski[4], Louise Thomsen Schmidt Arenholt[1,2]

1 Centre for Clinical Research, North Denmark Regional Hospital, Hjoerring, Denmark, 2 Department of Clinical Medicine, Aalborg University, Aalborg, Denmark, 3 Association K'OLO VANONA, Antananarivo, Madagascar, 4 Department of Health Science and Technology, Aalborg University, Aalborg, Denmark

* karl.joeker@rn.dk

## Abstract

Female genital schistosomiasis (FGS) is a neglected but highly prevalent disease in sub-Saharan Africa, caused by Schistosoma haematobium egg-induced inflammation in the pelvic region. FGS is characterized by four mucosal lesion types in the lower female genital tract: grainy sandy patches (GSP), homogeneous yellow sandy patches, abnormal blood vessels, and rubbery papules. This study focuses on the segmentation of cervical GSP lesions using a deep-learning convolutional neural network. A total of 583 cervical images from women in a S. haematobium endemic region of Madagascar, all exhibiting FGS-associated lesions, particularly GSP lesions, were used for this study. Weak annotations (non-pixel-wise) were generated using QubiFier software. A U-Net model with a focal loss function, and an Adam optimizer was trained to segment GSP lesions. A 5-fold cross validation was performed, thus resulting in 5 models. The models were evaluated on a dedicated test set, where model predictions were compared to the annotations. The average results of the models after cross validation were a DICE score of 0.61, accuracy of 0.81, sensitivity of 0.84, and specificity of 0.81. While the models performed well, the performance was affected by factors such as weak annotations, limited number of images, and image quality issues in the form of artifacts like specular reflections. These findings highlight the potential of U-Net-based models for automated lesion segmentation of FGS. Integration of such models into smartphone-based diagnostic tools could enable real-time detection and possible diagnosis of FGS in regions lacking specialized medical equipment or expertise. This approach may enhance access to early diagnosis, particularly in rural and underserved areas of sub-Saharan Africa, where FGS remains a significant public health burden. Future work should focus on enhancing model performance, validating using external datasets, and exploring

**Data availability statement:** The image data underlying the findings of this study consist of de-identified but sensitive clinical images obtained during gynaecological pelvic examinations. Although the images contain no direct identifiers, they are not publicly available due to ethical considerations related to participant privacy and the intimate nature of the data. The images are securely stored in a REDCap data management system hosted by the Centre for Clinical Research, North Denmark Regional Hospital, Denmark. Access to the de-identified images may be granted to qualified researchers upon reasonable request and with approval from the Centre for Clinical Research. Requests should be directed to forskning.rhn@rn.dk.

**Funding:** The study was supported by the Marie Pedersen and Jensine Heiberg Grant (968312) (KEJ). The funders had no role in study design, data collection and analysis, decision to publish, or preparation of the manuscript.

**Competing interests:** The authors have declared that no competing interests exist.

feasibility for mobile integration, offering a cost-effective solution for point-of-care FGS detection.

---

## Author summary

Female genital schistosomiasis (FGS) is a common but often overlooked disease affecting millions of women in sub-Saharan Africa. It is caused by a parasitic infection that leads to inflammation and damage in the female reproductive tract. One of the key signs of FGS is the presence of grainy sandy patches on the cervix, which are difficult to detect without specialized equipment and training. In this study, we explored the use of artificial intelligence to help identify these lesions in cervical images. We trained a deep learning model to recognize the grainy sandy patches using images collected from women in Madagascar, where the disease is widespread. Our model showed promising results, correctly identifying lesions in most cases. Although there were some challenges, such as image quality and limited annotation detail, our findings suggest that this technology could improve the accuracy of FGS diagnosis. In the future, it may be integrated into mobile devices to support diagnosis in rural clinics. By making detection easier and more accessible, we hope this approach can help improve early diagnosis and treatment of FGS, especially in areas with limited healthcare resources.

## Introduction

Female genital schistosomiasis (FGS) is a serious and often neglected disease that primarily impacts women and girls in Sub-Saharan Africa, where an estimated 56 million are affected [1]. FGS is caused by an infection from *Schistosoma haematobium* and, to a lesser extent, *Schistosoma mansoni,* both of which are transmitted through contact with fresh water infested by these parasites [2]. The condition arises when parasite eggs are deposited in the genital tissues, triggering acute and chronic inflammation together with tissue damage [3].

Efforts to diagnose FGS are substantially limited by the absence of an affordable gold-standard diagnostic method that reliably achieves high sensitivity and specificity. While biopsies revealing parasitic eggs in tissue is considered reference, biopsies are not routinely performed anymore due to their invasive nature and risk of HIV-acquisition post biopsy [4]. Other diagnostic approaches also present limitations. PCR testing for *Schistosoma* DNA requires significant laboratory infrastructure; circulating anodic antigen assays and egg counts in urine or feces are not specific to FGS; and Pap smears, for microscopy, have reported sensitivities lower than 15% [4]. According to the World Health Organization (WHO), the standard approach for diagnosing FGS is visual detection of FGS associated lesions in the lower genital tract during a pelvic exam, particularly the cervical surface and vaginal walls. The four

lesion types associated with FGS are grainy sandy patches (GSP), homogeneous yellow sandy patches, abnormal blood vessel formation, and rubbery papules [5]. For visual inspection during the pelvic exam, a digital camera or colposcope can be used [5]. However, diagnostic capacity is constrained by the high costs of equipment and the scarcity of specialized medical expertise, particularly in rural, resource-limited settings in Sub-Saharan Africa [6]. Additionally, visual diagnosis of FGS is challenging due to the presence of conditions such as sexually transmitted infections or cervical dysplasia, which can produce similar appearances and symptoms. Diagnostic accuracy is limited by the subjective interpretation of lesions and variability among experts and the subtle nature of FGS lesions. In a study by Sturt et al. [7], a comparison of two reviewers' assessments of lesions yielded a Cohen's kappa of 0.16, indicating only a "slight" agreement on the FGS diagnosis [7].

Addressing this challenge calls for the development of alternative diagnostic approaches that may improve FGS diagnostics in endemic areas and reduce subjectivity. Given that the current recommended approach is the detection of visual FGS and that FGS is characterized by specific lesion types visible through digital imaging, it is appropriate to explore computer vision and machine learning methods. Having models that can aid in the diagnosis of FGS opens possibilities for accessible, repeatable and objective point of care diagnosis, potentially through smartphone-based digital platforms in regions with limited access to specialist expertise or colposcopes.

As FGS remains an underrecognized tropical disease, only a few studies have investigated the application of computer vision for its diagnosis. Nevertheless, work by Holmen et al [8,9], has begun to explore this area. In one study, they employed colorimetric analysis to detect GSP and homogenous yellow sandy patches, the characteristic yellow lesions associated with FGS [8]. They developed an automatic region of interest cropping accompanied by an adaptive colorimetric algorithm that adjusts based on the mean brightness of the image. This approach achieved a sensitivity of 83% and a specificity of 73%. However, this technique is constrained because it relies purely on color data and lacks the capability to differentiate FGS-associated lesions from other possible conditions [8]. In another study, Holmen et al. utilized morphological analysis to identify blood vessels linked to FGS, achieving a sensitivity of 78% and a specificity of 80% [9]. The described methods are rule-based computer vision techniques, relying on predefined parameters or algorithms for detection.

Examining approaches from computer-aided diagnosis (CAD) of cervical cancer, a comparable medical challenge, it is evident that machine learning models are explored. A study by Jin et al. [10] reviewed 13 CAD methods for cervical cancer, focusing on their applicability to FGS by evaluating metrics such as sensitivity, specificity, and precision; notably 11 of the 39 methods utilized neural network algorithms. In the review, results from different studies achieved accuracies ranging from 68.90% to 98.55%, sensitivities from 57.56% to 99.49%, and specificities from 78.55% to 99.36% [10]. This review shows the potential for machine learning approaches and sets the scene for potential methods for FGS diagnosis.

Architectures such as ResNet have demonstrated strong performance in image-level classification [11,12] and Faster R-CNN have shown effectiveness in bounding-box detection [13], making these approaches promising candidates for exploring classification and detection methods for the diagnosis of FGS. However, these methods require a balanced dataset of cervical images with and without FGS lesions to achieve robust performance. Because the dataset available for this study contains only images with FGS lesions and lack healthy controls, this study focused on lesion segmentation to explore the potential of deep-learning methods for localizing FGS lesions in cervical images. In the available dataset, both healthy and infected tissue pixels are present within each of the images, enabling training of a segmentation model. Among available segmentation methods, convolutional autoencoder architectures like U-Net are well-suited for pixel-wise segmentation, enabling clear delineation of lesion areas [14]. In addition, U-Net effectively handles limited datasets through data augmentation techniques, making it a strong choice for this task, given the dataset consists of 583 images. For simplicity, this study focused on a single lesions type -GSP- as these lesions are widely prevalent, have been associated with biopsy-proven FGS in multiple studies and are not associated with other gynecological conditions [15–17].

The aim of this study was to develop a U-Net-based deep learning model designed for automated segmentation of GSP lesions. The potential future application of this methodology would allow for model integration in software for research,

colposcopes or other potential diagnostic devices, such as smartphones, subsequently enhancing FGS diagnosis in sub-Saharan Africa and other rural areas where FGS is prevalent, but access to clinical diagnosis is limited or impossible.

## Methods and materials

### Ethics statement

Ethical approval was obtained from the Ethics Committee at the Ministry of Health in Antananarivo (Authorization Number: 098-MSANP/CERBM; Number: 059-MSANP/CERBM; Number: 065MSANP/SG/-AGMED/CNPV/CERBM). The study was conducted following the principles of the Declaration of Helsinki and followed the Guidelines for Good Clinical Practice, as well as complied with local legislation and institutional requirements. Written informed consent was obtained from all participants, both adults and minors. Given that FGS can affect girls from pre-puberty and lead to life-threatening complications during adolescence, it was deemed unethical to exclude 15- to 17-year-old participants from the study. To maintain the confidentiality of these adolescent participants, parental or legal guardian consent was not sought, a procedure approved by the Ethics Committee

### Dataset

The images used in this study were obtained from a randomized controlled trial study that evaluated the efficacy and safety of a repeated-dosing regimen of the antiparasitic drug praziquantel in women with FGS, living in an S. haematobium endemic area in northwest Madagascar [18]. The original RCT study is registered on ClinicalTrials.gov under the identifier NCT04115062. The main study outcome was the evaluation of FGS-associated cervical lesions. The study included 116 women aged 15–35 years who presented with urogenital symptoms and had at least one FGS-associated cervical lesion observed during a pelvic examination. A speculum-based gynecological exam was performed, and images of the cervix were captured using a digital camera (Canon EOS M50). The camera was equipped with a 100mm macro-lens with a mounted circular LED light and a polarization filter. During capture, the camera was placed approximately 30 cm from the cervix on a tripod and microscope sledge, allowed precise distance control, and set to auto-focus mode. For each patient, the setup was adjusted to center the cervical orifice with a 3–5mm rim. Because of challenges in standardizing the imaging of vaginal lesions, only cervical lesions were documented and evaluated [18]. As part of the study the included women received either a standard single dose (40mg/kg) or a repeated dose regimen at baseline, as well as additional praziquantel treatment at follow-up visits. The cervical images were obtained at baseline, and then after 5 weeks, 10 weeks, 15 weeks, 12 months, and 18 months. The best quality image from each woman was selected and stored in servers hosted at the North Denmark Regional Hospital [18]. A total of 590 cervical images from the 6 visits of the 116 women were available for analysis following loss to follow-up and quality assessment through consensus between LTA and KJ. Images were considered of insufficient quality when the cervix was out of focus or not fully visible.

### Annotation with QubiFier

The images obtained from the RCT study by Arenholt et al. [18] were annotated using QubiFier, a validated digital gridded imaging software, by a team of three trained medical students who were supervised by a consultant gynecologist with extensive experience in assessing FGS-related lesions (LTA) [19,20]. In each full-resolution image, a superimposed grid consisting of 424 equally sized squares was placed so that the entire surface of the cervix was covered. This grid served as the region of interest (ROI). For the annotation, each square was manually marked if a cervical lesion was present. An example of this can be seen in Fig 1. For detailed information on the annotation process, see [19]. However, these grid-based annotations introduce some ambiguity due to their lack of pixel-level precision. While less accurate than pixel-wise labeling, generating precise annotations would be costly and time-consuming. Given that the dataset had already been reviewed by FGS experts using QubiFier, the grid-based approach was deemed the most practical solution for this study

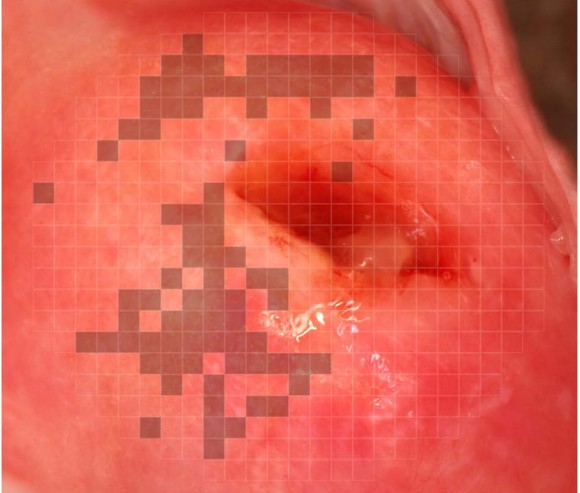

**Fig 1. QubiFier labeling.** A cervical image is evaluated for GSP lesions using QubiFier software [20]. The shaded areas on the grid are manually marked by an expert in the QubiFier software. Raw images sourced from https://doi.org/10.3389/fitd.2024.1322652.

## Preprocessing

Before training the models, the data was preprocessed in multiple ways. First, labels were created from the annotations made in QuibFier, then specular reflections (SRs) were filtered on the raw training images. Finally, the ROI was imposed on both labels and input images

**Label creation.** Since QubiFier was not originally designed to generate labels for machine learning, specific image processing steps were required to create binary image labels for the U-Net model. These labels were derived through affine registration, image differencing, filtering, and morphological operations, resulting in binary versions. During the label creation, 7 images could not be aligned properly with the given QubiFier annotation, therefore these image-sets were discarded. The remaining dataset consisted of 583 images after label creation. For visualization of the label creation process refer to the S1 and S2 Figs.

**Removal of specular reflection.** SRs appear as intense white spots in cervical images due to the glossy and moist nature of cervical tissue. SRs act as noise for the model, so removing them involves two key steps: (1) identifying and masking the SRs, and (2) inpainting the mask to cover the reflections, an approach adapted from Nie et al.'s work on SR removal in endoscopic images [21]. The reason for inpainting the reflections instead of disregarding these areas for training is because of the limited available data. The objective was to obtain as much information from the images as possible. For detailed information about the localization of specular reflections, see S1 Appendix.

Following the localization of SRs, Navier-Stokes inpainting from OpenCV [22] was utilized to eliminate them. Fig 2 illustrates the detection and removal process: Fig 2a is the original image, Fig 2b highlights the SRs, and Fig 2c presents the inpainted outcome.

**ROI handling.** To ensure that the models only trained within the parameters of the annotations, a ROI was imposed on both label images and the input images. This ROI was a mask with the shape of the grid from the QubiFier software. This ensured that everything outside this ROI was not taken into account when training the models. After preprocessing and labeling, the dataset included 583 images with matching labels. The reduction in images resulted from the process of creating labels from the QubiFier annotations, see label creation section.

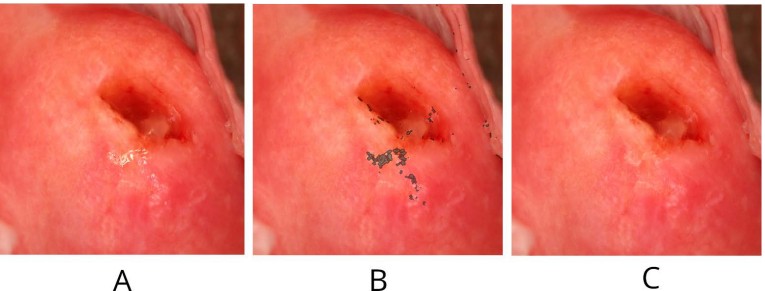

|   A   |   B   |   C   |

**Fig 2. Visualization of steps for specular reflection removal. (a)** Displays the raw image still including the SR. **(b)** Displays the image with the SR marked with darker pixels. **(c)** Displays the image after SRs have been removed using an inpainting technique to cover the SRs using similar pixel values as the surrounding pixels. Raw images sourced from https://doi.org/10.3389/fitd.2024.1322652.

## Model architecture

The architecture used for the model is based on the U-net model [14], which is designed for effective performance on limited datasets, a common scenario in biomedical imaging. U-net combines a contracting path to capture context and an expanding path for precise localization, enabling detailed segmentation tasks.

The model in this study is similar to the original U-net, with the exception of padding. Padding is used after each convolutional layer, ensuring there is no reduction in output dimensions. The padding ensures consistent image dimensions from input to output, preserving spatial information at the boundaries. Fig 3 illustrates the model structure.

## Model training

The hardware used to train the model was a 2022 Dell XPS 15 9520 equipped with an NVIDIA GeForce RTX 3050 Ti Mobile GPU (4 GB VRAM), a 12th-generation Intel Core i7-12700H processor, and 16 GB of system memory. Although this configuration was not ideal for extensive deep-learning experimentation, it provided sufficient computational capacity to train the model on the present dataset within a reasonable time frame.

**Dataset split for training.** For training, the dataset was split at participant-level to ensure that participant images in training were not also present in the validation or test set. The dataset aimed to use 70% of the participants for training, 15% for validation, and 15% for testing. This resulted in 388 images for training, 97 for validation, and 98 for testing. To generalize better on the given data, 5-fold cross validation was employed. The data split was done 5 different times where different participants were held out for validation when training. The participants held out for the final testing remained the same for each fold.

**Training parameters.** The dataset exhibits class imbalance, as in some samples the GSP annotations correspond to relatively small parts of ROI, resulting in a disproportionate ratio between foreground and background pixels. To mitigate the effects of this imbalance, training parameters were selected to improve model performance.

The models were trained using the Adam optimizer with a learning rate of 0.001 and the focal loss function with an alpha parameter value of 0.25 and a gamma parameter value of 3. Focal loss was chosen due to the imbalance of the dataset. All images were resized to 512x512 pixels, data augmentation was applied to enhance the model's generalization capabilities. These augmentations were random horizontal flip, random vertical flip, and random rotation. The models were trained with a batch size of 32 and for 50 epochs, which means they train on the dataset 50 times. To avoid overfitting, early stopping was employed during training. This meant that if the validation loss did not reach a new minimum for 10 epochs, training was interrupted and the model with the lowest validation loss was considered the best. The learning rate, focal loss alpha, and gamma, were selected empirically based on previous experimentation. More information about the parameter selection can be found in S1 Appendix and S1 Table.

PLOS Neglected Tropical Diseases

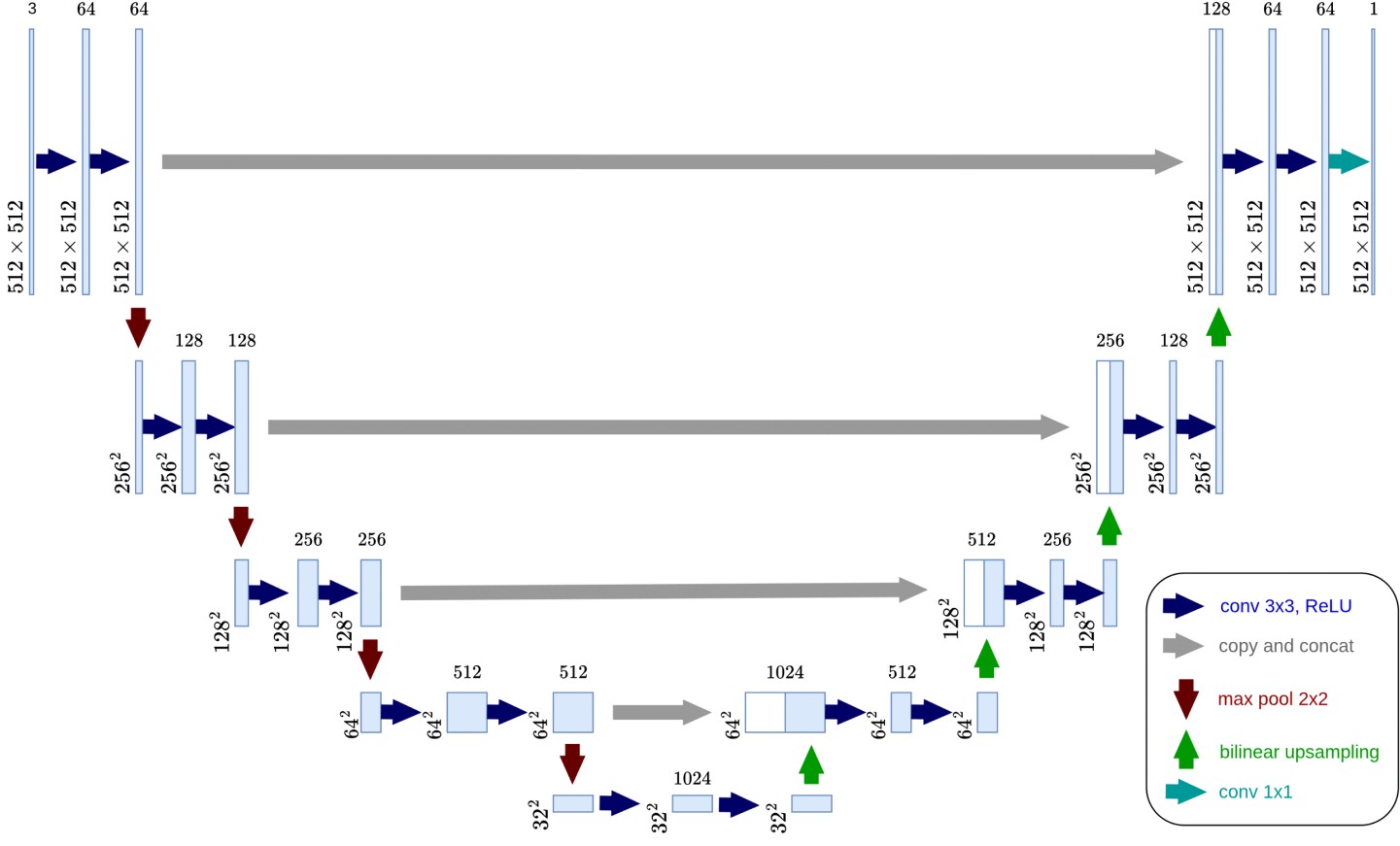

**Fig 3. Model Architecture.** Model based on the U-NET architecture from Ronneberger et al. [14].

## Model evaluation

To evaluate the models, accuracy, sensitivity, specificity, the receiver operator characteristics (ROC) curve, and the DICE score were calculated, each reflecting different facets of the model's performance. Accuracy provides a broad overview but has limitations with imbalanced data, so it is not used alone for assessment. Sensitivity is the measure of the true positives, and in segmentation, sensitivity evaluates the predicted foreground. Conversely, specificity measures true negatives among the negatively classified, indicating the predicted background in segmentation. The ROC curve visually displays the trade-off between the True Positive Rate (TPR) and False Positive Rate (FPR) across different thresholds, gauging at the model's class discrimination capability. A curve near the top-left corner signifies a high TPR with few false positives. The best threshold is at the point nearest to the upper-left corner. The DICE score, calculated as:

$$DICE = \frac{2|A \cap B|}{|A| + |B|}$$

(1)

Where $A$ is the ground truth and $B$ is the predicted segmentation, ranging from 0 to 1, with values closer to 1 indicating better performance. These evaluation metrics provide multiple insights into the model's capabilities, with the DICE score in particular measuring the overlap between prediction and annotated truth. For each of the 5 models trained, the ROC curve was calculated to obtain the optimal threshold for segmentation as well as area under the curve (AUC), which indicates pixelwise classification across thresholds.

## Results

The following ROC curves were calculated shown in Fig 4. The models reached an average AUC of 0.89, indicating a relatively strong pixel-wise classification performance.

Table 1 presents the models' quantitative evaluation metrics the five different models from cross validation as well as the average across the models. The metrics include the ROC AUC, the optimal threshold, DICE score, accuracy, sensitivity, and specificity. The models demonstrated moderate segmentation performance, with an average DICE score of 0.61 and an average sensitivity of 0.94. Additionally, the models achieved an average accuracy of 0.81 and an average specificity of 0.81, indicating a somewhat well-balanced pixel classification with effective false positive and false negative control.

While the quantitative metrics provide a general understanding of the model's performance, qualitative analysis through visualizations offers deeper insights into the segmentation behavior. Fig 5 shows three examples of predictions by the first

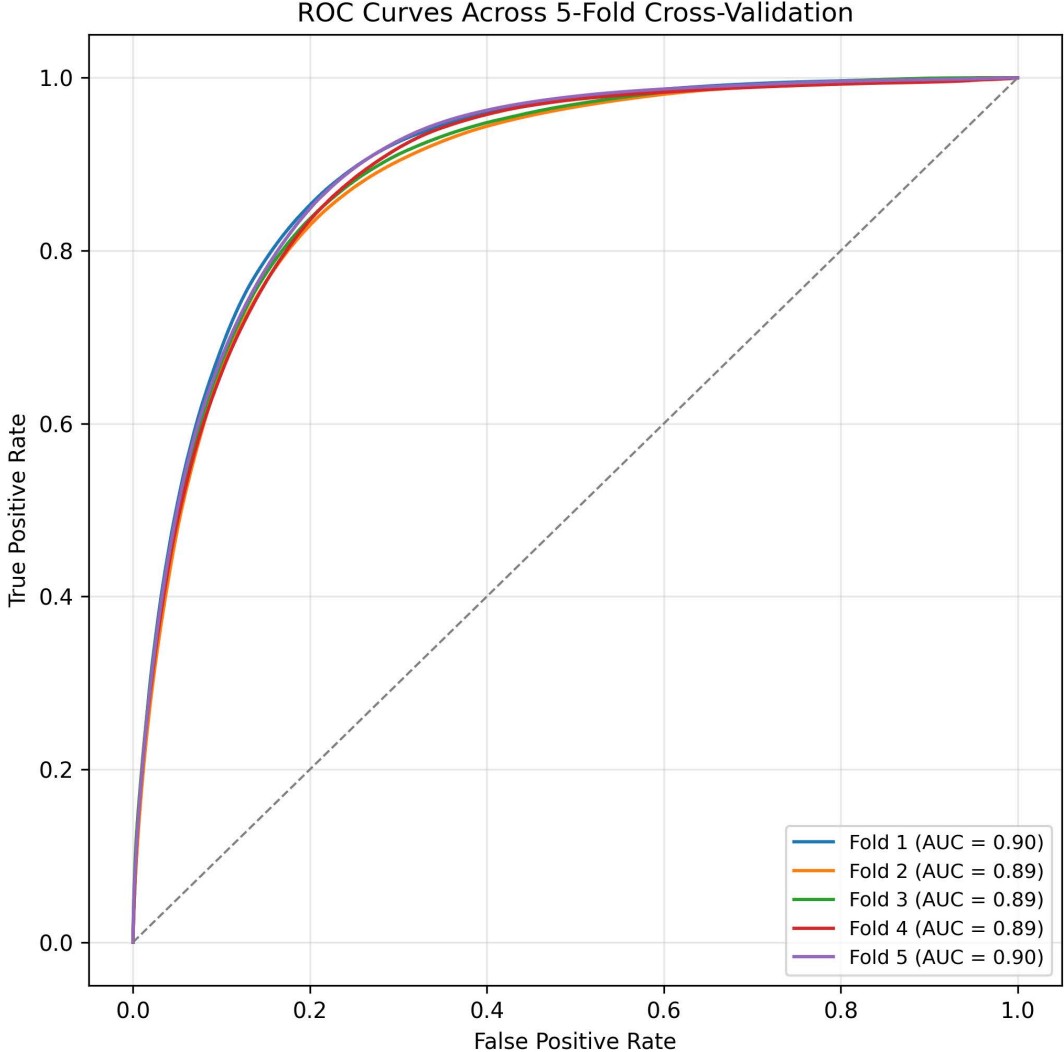

**Fig 4. Receiver Operator Characteristics (ROC) Curve.** The ROC curve for all models of the 5-fold cross validation.

**Table 1. Average performance metrics on the test set.**

| Model fold | 1 | 2 | 3 | 4 | 5 | Cross validation average |
|---|---|---|---|---|---|---|
| ROC AUC | 0.9 | 0.89 | 0.89 | 0.89 | 0.9 | 0.89 |
| DICE | 0.62 | 0.6 | 0.6 | 0.6 | 0.61 | 0.61 |
| Accuracy | 0.82 | 0.81 | 0.81 | 0.81 | 0.81 | 0.81 |
| Sensitivity | 0.84 | 0.83 | 0.83 | 0.84 | 0.84 | 0.84 |
| Specificity | 0.81 | 0.81 | 0.81 | 0.8 | 0.81 | 0.81 |

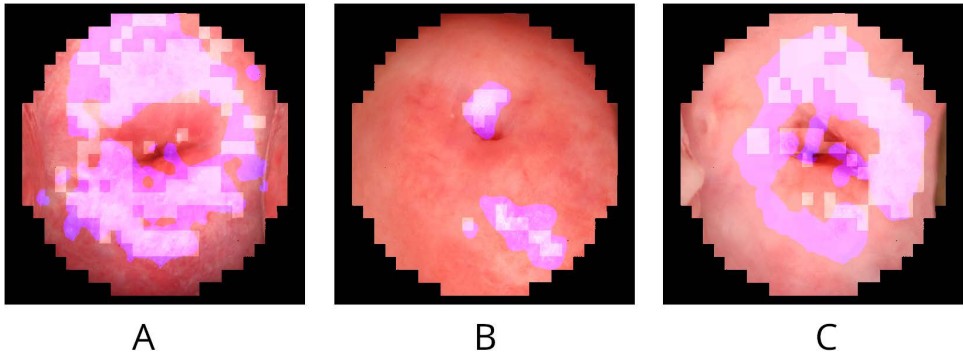

A          B          C

**Fig 5. Model segmentation visualization.** (a) and (b) visualize a good overlap between prediction and annotation, while (c) visualizes moderate over-lap. White grid cells represent the masked labels, the purple markings indicate the predicted pixels, the black represent pixels outside the ROI. Annotated images sourced from: https://doi.org/10.3389/fitd.2024.1322652.

model, with the best metrics, on the test set. The model exhibited some degree of congruence with the annotated truth labels. Fig 5a and 5b illustrates a scenario where the model's predictions are aligned effectively with the label. Conversely, Fig 5c depicts instances where the model's prediction aligned with the label. However, in these instances, the model appeared to over-segment the image, which, while enhancing sensitivity, diminished overall accuracy and specificity. The suboptimal segmentation could be attributed to residual artifacts resulting from specular reflection inpainting. Overall, the models demonstrated moderate performance yet face some challenges.

## Discussion

The models achieved an average AUC of 0.89, indicating a high level of discriminative power in distinguishing pixels with GSP lesions from those without. A sensitivity of 0.84 suggests that the model successfully identified the majority of lesion-positive pixels, while a specificity and accuracy of 0.81 reflect a balanced performance that minimizes both false positives and false negatives. This level of accuracy is promising for use in screening settings, particularly in low-resource contexts where expert diagnosis is unavailable. The model's segmentation performance, with a DICE score of 0.61, reflects a moderate ability to localize lesions at the pixel level. This outcome likely stems from the visual variability and subtlety of GSP lesions, combined with the presence of imaging artifacts such as residual specular reflections after preprocessing, weak labeling, and the limited dataset size. While this score indicates the model can identify general lesion regions, it may fall short in applications requiring precise lesion boundaries, such as monitoring lesion progression or calculating precise lesion proportions.

Compared to other computer vision approaches by Holmen et al. [8,9], the results highlight several advantages of the U-Net-based approach. In one study, they achieved a sensitivity of 0.83 and a specificity of 0.73 using a colorimetric method to identify GSP and homogenous yellow sandy patches [8]. However, because their method relied solely on color

features, it may be less robust to variation in lesion appearance. Similarly, their morphological analysis [9] of abnormal blood vessels yielded a sensitivity of 0.78 and specificity of 0.80, revealing limitations in precision. In contrast, this study's U-Net model achieved higher sensitivity (0.84) along with balanced accuracy and specificity (both 0.81). While direct comparisons must be interpreted cautiously due to differences in datasets and evaluation criteria, the higher sensitivity and balanced metrics in this study suggest improved performance, which may be attributable to the model's ability to learn hierarchical and contextual features.

The review study by Jin et al. [10] focusing on deep learning methods for computer aided diagnostics of cervical cancer demonstrated the potential of deep learning in automated detection and classification, providing a strong foundation for the application of these techniques in medical imaging. Although their methods focused on classification, the underlying principles of deep learning for image analysis are applicable to lesion segmentation in FGS. By leveraging deep learning architectures like ResNet and support vector machines, these prior works highlight the promise of automated solutions in gynecological diagnostics. The U-Net model used in this study builds on these advances, adapting them for the pixel-wise segmentation of GSP lesions, offering a promising tool for FGS diagnosis in resource-limited settings.

## Limitations

Several challenges impacted the model's performance and highlighted areas for improvement. First, the dataset's weak annotations, while practical and a cost-effective alternative to pixel-wise labeling, introduced a degree of uncertainty that likely constrained segmentation precision, as reflected in the moderate DICE score (0.61). Second, in the visual results of the model, the residual artifacts visible after preprocessing with SR removal indicate the need for increased robustness in the SR removal algorithm. Third, the reliance on a relatively small dataset may have hindered the model's ability to generalize across diverse patient populations and imaging conditions. Addressing these data quality and quantity issues is critical for improving sensitivity and lesion delineation.

An additional limitation of this study is the lack of external validation and negative control evaluation. The model was trained and tested exclusively on a single dataset, and its performance was not assessed on an independent external cohort. As a result, the generalizability of the reported results to other institutions, imaging protocols, or patient populations remains uncertain. Furthermore, the dataset did not include FGS-negative images, preventing evaluation of the model's behavior in the absence of target lesions. Without such negative controls, it is unclear whether the model might erroneously segment GSP-like structures in FGS-negative images, potentially leading to false-positive detections. Future work should incorporate external datasets and FGS-negative cases to more rigorously assess specificity, robustness, and clinical applicability.

While this study explored the dataset's potential for supporting segmentation of GSP lesions, the moderate DICE score suggests that the available data may not be clean or consistent enough to enable precise lesion delineation. This outcome raises the possibility that a classification-focused model may be more appropriate as a next step. Given the relatively high sensitivity (0.84) achieved in this study, future work could leverage this strength to develop a classification model that reliably identifies lesion-positive cases, potentially offering a more robust and data-efficient diagnostic tool in resource-limited settings.

## Perspectives

Several avenues for future research are recommended to improve the robustness and applicability of this approach. First, developing larger and more diverse datasets with higher-quality annotations, potentially through semi-automated labeling strategies for efficiency, would significantly enhance model performance. Second, exploring alternative architectures could provide additional gains in segmentation accuracy and sensitivity. Third, incorporating more robust preprocessing methods to mitigate image artifacts, such as specular reflections, would address the key limitations observed in this study.

Potential future applications after further improvement of this model include the integration of mobile phone-based tools for point-of-care diagnosis in resource-limited settings, with significant implications for rural regions, particularly in Sub-Saharan Africa. By relying on automated visual detection, this approach reduces the need for specialized clinical expertise, limits the use of diagnostic procedures that require tissue sampling or laboratory infrastructure, and enables on-site assessment with mobile tools, offering a scalable and cost-effective solution for early detection and management of FGS. These methods can be expanded to other FGS lesion types, becoming an all-encompassing FGS diagnostic tool. Future work should focus on enhancing data quality, exploring advanced architectures, and validating the models with external data sets, to ensure reliability in edge cases.

## Conclusion

The study shows promise in applying a deep learning model with a U-net architecture. The model effectively identifies GSP lesions but has some limitations. Further improvements are needed for better segmentation of subtle features. The results highlight the potential of using deep learning models for automated diagnostics. This study represents a crucial step toward integrating artificial intelligence into healthcare solutions for neglected diseases, with the potential to improve diagnostic accuracy, accessibility, and patient outcomes in underserved populations.

## Supporting information

**S1 Fig. Affine registration and cropping.** The annotation image is aligned with the original, unannotated image. To perform image differencing, both images must be the same size and registered in the same coordinate space. Registration is achieved by matching Scaled Invariant Feature Transforms and using Random Sample Consensus to find the best transformation matrix between the two images. The original image is then cropped to match the dimensions of the cropped annotation. Created using source images from [19].
(TIF)

**S2 Fig. Binary image mask.** The annotation mask is extracted by computing the pixel-wise difference between them to isolate annotated regions. The difference image is converted to grayscale and filtered with a median filter to reduce noise. Afterward, binarization at a threshold of 0.5 creates a binary mask, and morphological opening and closing operations are applied to remove residual noise and fill small gaps. The resulting mask is a clean binary representation of the annotations. Created using source images from [19].
(TIF)

**S1 Appendix. Specular reflection methodology and Hyper-parameter tuning from previous experimentation [21,23,24].**
(DOCX)

**S1 Table. Hyperparameter search space.** The batch size was fixed at 32, and the number of epochs was set to 100. These choices were influenced by the limited size of the dataset and the augmentations applied to balance the data and improve generalization. After running the grid search, the model with the highest average intersection over union, a calculation representing the model's overlapping capability, across all epochs, was selected as the best-performing model. Once the model was selected, the validation loss for each epoch was analyzed, and the epoch with the lowest validation loss was selected as the model with the best generalization.
(DOCX)

## Author contributions

**Conceptualization:** Karl Emil Jøker, Peter Christian Derek Leutscher, Kristine Brøndbjerg Øby, Karoline Jøker, Louise Thomsen Schmidt Arenholt.

**Data curation:** Karoline Jøker, Louise Thomsen Schmidt Arenholt.

**Formal analysis:** Karl Emil Jøker, Karoline Jøker, Louise Thomsen Schmidt Arenholt.

**Funding acquisition:** Karl Emil Jøker, Peter Christian Derek Leutscher.

**Investigation:** Karl Emil Jøker, Peter Christian Derek Leutscher, Kristine Brøndbjerg Øby, Karoline Jøker.

**Methodology:** Karl Emil Jøker, Peter Christian Derek Leutscher, Maciej Plocharski.

**Project administration:** Karl Emil Jøker, Peter Christian Derek Leutscher.

**Resources:** Kristine Brøndbjerg Øby, Karoline Jøker, Bodo Sahondra Randrianasolo, Louise Thomsen Schmidt Arenholt.

**Software:** Karl Emil Jøker, Kristine Brøndbjerg Øby.

**Supervision:** Peter Christian Derek Leutscher, Maciej Plocharski, Louise Thomsen Schmidt Arenholt.

**Validation:** Karl Emil Jøker.

**Visualization:** Karl Emil Jøker.

**Writing – original draft:** Karl Emil Jøker, Maciej Plocharski.

**Writing – review & editing:** Karl Emil Jøker, Peter Christian Derek Leutscher, Karoline Jøker, Maciej Plocharski, Louise Thomsen Schmidt Arenholt.

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
