## [Decision Letter · Decision Letter 0]

5 Nov 2025

associated with female genital schistosomiasis using deep

Response to Reviewers
Revised Manuscript with Track Changes
Manuscript

Shaden Kamhawi

co-Editor-in-Chief

Paul Brindley

co-Editor-in-Chief

**Additional Editor Comments:**

Introduction – while the description of similar models in use for cervical cancer sets the scene and utility of these methods for FGS, the introduction currently reads more like a review. Specific context is needed regarding FGS and the current diagnostic methods, including sensitivity and specificity.

Line 92 – given the loss of images from inadequate image quality, can the authors provide further details regarding what is meant by ‘standarized manner using a digital camera’? What was the protocol to ensure comparable images?

Line 99 – what level of image resolution is used for this the QubiFier? How do the FGS experts grade the lesions – this information could be included in the introduction?

**Journal Requirements:**

3) Some material included in your submission may be copyrighted. According to PLOSu2019s copyright policy, authors who use figures or other material (e.g., graphics, clipart, maps) from another author or copyright holder must demonstrate or obtain permission to publish this material under the Creative Commons Attribution 4.0 International (CC BY 4.0) License used by PLOS journals. Please closely review the details of PLOSu2019s copyright requirements here: PLOS Licenses and Copyright. If you need to request permissions from a copyright holder, you may use PLOS's Copyright Content Permission form.

Potential Copyright Issues:

- Please confirm (a) that you are the photographer of Figures 1, 2, 5, S1, and S2., or (b) provide written permission from the photographer to publish the photo(s) under our CC BY 4.0 license.

**Reviewers' comments:**

**Key Review Criteria Required for Acceptance?**

**Methods**

-Are the objectives of the study clearly articulated with a clear testable hypothesis stated?

-Is the study design appropriate to address the stated objectives?

-Is the population clearly described and appropriate for the hypothesis being tested?

-Is the sample size sufficient to ensure adequate power to address the hypothesis being tested?

-Were correct statistical analysis used to support conclusions?

-Are there concerns about ethical or regulatory requirements being met?

Reviewer #1: What steps were taken to avoid overfitting? As this is a relatively small dataset, some comments on how the risk of overfitting were addressed would be appropriate. The relatively small testing set (n = 87) should also be addressed when discussing the results.

L99-108: Please could the authors comment on the number of reviewers, their qualifications and training to label images? Or specifically refer the reader to the relevant paper? These sorts of details are useful when determining the accuracy of the ground truth annotations.

L98-108: How was the region of interest defined? If your model only focuses on the cervix, and not the surrounding tissue, then this will need to be specified.

L169-170: Your data was longitudinal, with multiple images per woman. Please can the authors comment on how they ensured that images from one woman were not used across both training and testing datasets, as this may create an uplift in model performance. Furthermore, do the authors foresee any risk of bias by having multiple timepoints per woman? Perhaps a comment (and the small amount of evidence we have on lesion progression) on the likelihood of lesions changing over the study period might be warranted?

L267-284: The model has not been run using any external datasets (understandably), which needs to be mentioned. Furthermore, it does not appear that the model has been run on images that do not have FGS lesions? Without negative controls, how do we know that the model won’t segment ‘GSP-like’ areas in FGS-negative images?

.

**Results**

-Does the analysis presented match the analysis plan?

-Are the results clearly and completely presented?

-Are the figures (Tables, Images) of sufficient quality for clarity?

Reviewer #1: (No Response)

**Conclusions**

-Are the conclusions supported by the data presented?

-Are the limitations of analysis clearly described?

-Do the authors discuss how these data can be helpful to advance our understanding of the topic under study?

-Is public health relevance addressed?

Reviewer #1: There are multiple mentions of integration into smartphones/mobile phones. L78-82 suggest that this is the only potential application of this work. However, the types of smartphones that are available and affordable in endemic low-resource settings may not always have the magnification and imaging capabilities to detect GSP lesions. I would encourage the authors to consider that there may be other ways of integrating this technology.

**Editorial and Data Presentation Modifications?**

Reviewer #1: More information on the types of lesions seen in the dataset is warranted. The abstract states that all 583 cervical images exhibited FGS-associated lesions. Please could the authors clarify whether this was any lesion type or GSP specifically? How many images had other lesion types and did the model ever segment these lesions instead?

More information is needed on the level of imbalance in the dataset. GSP generally only take up a tiny percentage of the overall ROI. This is important for assessing class imbalance and for understanding the use of a focal loss function (and the degree of alpha and gamma used).

L83-108: Some more detail on the imaging, annotation and classification process is needed. These details are essential when considering how successfully a model may be able to perform and generalise.

L91: Some brief details on the ‘standardized manner’ would be beneficial.

L96: Please could the authors comment on how many images were discarded due to ‘inadequate image quality? These were separate from the 7 lost during preprocessing? What was considered poor quality (e.g. just blurring? ROI not in view? Discharge in view?)?

L119-153: The section detailing the removal of SRs has a large amount of detail compared to other methods sections that may be more relevant to the objective of the paper (and of more interest to a PLOSNTD audience). Much of the detail could be added to a supplementary file to free up space.

L151-152: Some more information about the kinds of anomalies and outliers would be useful.

L166-167: The author could provide more detail on the type of hardware used so the reader could appreciate the limitations fully.

L238-239: The mention of visual variability and subtlety here is good. However, think should be mentioned earlier, before the discussion.

**Summary and General Comments**

Reviewer #1: This model focuses on one of the four FGS lesion types. This is likely to be a significant limitation of the model in terms of its diagnostic usefulness for FGS. GSPs are not often the most common lesion type reported in the literature. It is understandable why the authors may have selected GSP as a starting point for the model but this needs to be explained, with this key limitation explained early. Careful distinction is needed when discussing FGS lesions in general, not GSP specifically.

Abstract: Please could the authors clarify what they mean by non-invasive? Even with a smartphone, the procedure would still require a speculum exam, which many women consider highly invasive. This is also mentioned in L296 and in the conclusion. Visual diagnostics, by their nature, will always be invasive.

Abstract: The use of the term “distinct mucosal lesions” would suggest that they are distinctive from other cervical mucosal changes, which is often not the case. Suggest rewording. Mention of ‘confounding cervical diseases’ (diseases that appear visually similar to FGS) would also be beneficial.

L15-17: Diagnostic capacity is also constrained by the high levels of subjectivity. I would suggest adding this into your introduction, as reducing subjectivity is a key factor in the benefits of computer vision.

L27: Sigve’s work is not recent. This paper was published ten years ago. Since then, there have been significant advancements computer vision (and FGS research in general).

PLOS authors have the option to publish the peer review history of their article (what does this mean? ). If published, this will include your full peer review and any attached files.

**Do you want your identity to be public for this peer review?** For information about this choice, including consent withdrawal, please see our Privacy Policy .

Reviewer #1: **Yes:** Morgan E. Lemin

**Figure resubmission:**

**Reproducibility:** To enhance the reproducibility of your results, we recommend that authors of applicable studies deposit laboratory protocols in protocols.io, where a protocol can be assigned its own identifier (DOI) such that it can be cited independently in the future. Additionally, PLOS ONE offers an option to publish peer-reviewed clinical study protocols. Read more information on sharing protocols at https://plos.org/protocols?utm_medium=editorial-email&utm_source=authorletters&utm_campaign=protocols

---

## [Editor Report · Decision Letter 1]

13 Feb 2026

Dear Mr Jøker,

We are pleased to inform you that your manuscript 'Image segmentation of cervical grainy sandy patches lesions

associated with female genital schistosomiasis using deep

convolutional neural network with U-NET architecture' has been provisionally accepted for publication in PLOS Neglected Tropical Diseases.

Best regards,

Krystyna Cwiklinski, PhD

Section Editor

Krystyna Cwiklinski

Section Editor

Shaden Kamhawi

co-Editor-in-Chief

Paul Brindley

co-Editor-in-Chief

The authors have addressed the comments raised during the review process. The manuscript is now suitable for publication in PLoS NTD.

---

## [Editor Report · Acceptance letter]

Dear Mr Jøker,

We are delighted to inform you that your manuscript, "Image segmentation of cervical grainy sandy patches lesions

associated with female genital schistosomiasis using deep

convolutional neural network with U-NET architecture," has been formally accepted for publication in PLOS Neglected Tropical Diseases.

Best regards,

Shaden Kamhawi

co-Editor-in-Chief

Paul Brindley

co-Editor-in-Chief
